# Expanding the Spectrum of Pancreatic Cancers Responsive to Vesicular Stomatitis Virus-Based Oncolytic Virotherapy: Challenges and Solutions

**DOI:** 10.3390/cancers13051171

**Published:** 2021-03-09

**Authors:** Molly C. Holbrook, Dakota W. Goad, Valery Z. Grdzelishvili

**Affiliations:** Department of Biological Sciences, University of North Carolina at Charlotte, Charlotte, NC 28223, USA; mpenton@uncc.edu (M.C.H.); dgoad@uncc.edu (D.W.G.)

**Keywords:** oncolytic virus, virotherapy, pancreatic cancer, pancreatic ductal adenocarcinoma, vesicular stomatitis virus

## Abstract

**Simple Summary:**

Pancreatic ductal adenocarcinoma (PDAC) is a devastating malignancy with a poor prognosis and a dismal survival rate. Oncolytic virus (OV) is an anticancer approach that utilizes replication-competent viruses to preferentially infect and kill tumor cells. Vesicular stomatitis virus (VSV), one such OV, is already in several phase I clinical trials against different malignancies. VSV-based recombinant viruses are effective OVs against a majority of tested PDAC cell lines. However, some PDAC cell lines are resistant to VSV. This review discusses multiple mechanisms responsible for the resistance of some PDACs to VSV-based OV therapy, as well multiple rational approaches to enhance permissiveness of PDACs to VSV and expand the spectrum of PDACs responsive to VSV-based oncolytic virotherapy.

**Abstract:**

Pancreatic ductal adenocarcinoma (PDAC) is a devastating malignancy with poor prognosis and a dismal survival rate, expected to become the second leading cause of cancer-related deaths in the United States. Oncolytic virus (OV) is an anticancer approach that utilizes replication-competent viruses to preferentially infect and kill tumor cells. Vesicular stomatitis virus (VSV), one such OV, is already in several phase I clinical trials against different malignancies. VSV-based recombinant viruses are effective OVs against a majority of tested PDAC cell lines. However, some PDAC cell lines are resistant to VSV. Upregulated type I IFN signaling and constitutive expression of a subset of interferon-simulated genes (ISGs) play a major role in such resistance, while other mechanisms, such as inefficient viral attachment and resistance to VSV-mediated apoptosis, also play a role in some PDACs. Several alternative approaches have been shown to break the resistance of PDACs to VSV without compromising VSV oncoselectivity, including (i) combinations of VSV with JAK1/2 inhibitors (such as ruxolitinib); (ii) triple combinations of VSV with ruxolitinib and polycations improving both VSV replication and attachment; (iii) combinations of VSV with chemotherapeutic drugs (such as paclitaxel) arresting cells in the G2/M phase; (iv) arming VSV with p53 transgenes; (v) directed evolution approach producing more effective OVs. The latter study demonstrated impressive long-term genomic stability of complex VSV recombinants encoding large transgenes, supporting further clinical development of VSV as safe therapeutics for PDAC.

## 1. Introduction

Pancreatic ductal adenocarcinoma (PDAC) is the most common form of pancreatic neoplasm. It is a highly invasive malignancy, which forms a stromal desmoplastic reaction (desmoplasia), characterized by a dramatic increase in the proliferation of alpha-smooth muscle actin-positive fibroblasts and an increased production of many extracellular matrix components [1]. Family history, diabetes, and smoking are the most well-established risk factors for developing pancreatic cancer. Despite being only the 13th most common type of cancer, PDAC is the fourth-leading cause of cancer-related deaths and is predicted to become the second-leading cause of cancer-related death by 2030, as incidence increases while rates of survivorship remain stagnant due to late diagnosis and limited treatment options [2].

*KRAS*, *CDKN2A*, *TP53*, and *SMAD4* serve as driver genes for PDAC development, and the vast majority of patients with fully established pancreatic cancer carry genetic defects in at least one of these genes [3]. Mutations in KRAS are present in 90% of PDAC tumors, 95% of PDAC tumors have mutations in *CDKN2A* (encodes p16), 50–75% in *TP53*, and *SMAD4* (*DPC4*) is lost in approximately 50% of PDAC tumors [4]. Mutated *KRAS* oncogene leads to an abnormal, constitutively active, Ras protein. This results in aberrant activation of pathways responsible for survival and proliferation [5]. Inactivation of the tumor suppressor gene *CDKN2A* results in the loss of p16, a protein that serves as a regulator of the G1-S checkpoint of the cell cycle. Abnormalities in *TP53* prevent it from acting as a tumor suppressor protein, including its important role as a regulator of DNA-damage checkpoints. Furthermore, many p53 mutants acquire devastating gain-of-function oncogenic activities, actually promoting cell survival, proliferation, invasion, migration, chemoresistance, and chronic inflammation. *SMAD4* (*DPC4*) is related to the TGF-β signaling pathway, but some mutations result in abnormal signaling by TGF-β, a transforming growth factor receptor on the cell surface which can further increase the risk of cancer development by increasing the rate of cell growth and replication. In addition, germline mutations within BRCA2, BRCA1, ATM, and other genes were frequently identified in PDACs as inherited traits increasing susceptibility to PDAC development later in life [6,7]. These genes, especially when identified as being comorbid, are correlated with a significantly higher metastatic burden [8,9,10].

The primary treatments for PDAC include surgery, chemotherapy, radiotherapy, and palliative care [11]. Surgical resection still retains the greatest chance of success for potentially curing PDAC, however late-stage diagnosis due to ambiguous symptoms often results in tumors that are too-far progressed for surgery alone. Less than 25% of patients that present with PDAC are eligible for surgical resection, and 5-year survivorship of completely resected patients is approximately 37% [4]. In addition, even in patients where surgical resection was performed with either preparatory or subsequent adjuvant chemotherapy, there is a high rate of recurrence, and up to 80% of patients with recurrent PDAC will relapse with local and/or distant disease, which is associated with mortality within 2 years from diagnosis.

Recent advances in the understanding of the molecular biology, diagnosis, and staging of PDAC will hopefully lead to greater progress in the development of novel treatment approaches for PDAC patients. One such approach is oncolytic virus (OV) therapy, which utilizes replication-competent viruses to preferentially infect, replicate in, and kill cancer cells. In this review, we will discuss current advances with OV therapy for PDAC, with a special focus on vesicular stomatitis virus (VSV), the major interest of our laboratory. For comprehensive reviews of gene therapy for pancreatic cancer (unlike oncolytic virotherapy, gene therapy is typically based on replication-defective viral vectors for transgene delivery), we refer to these excellent papers [12,13].

## 2. Major Challenges with Current PDAC Treatments

Since 1997, gemcitabine-based chemotherapy has been the standard first-line treatment for patients with unresectable locally advanced, or metastatic pancreatic cancer with a median survival rate of 4.4–5.6 months, especially when patients are not healthy enough for combination therapies [14]. Gemcitabine (dFdC) is an analog of deoxycytidine and a pro-drug that, once transported into the cell, must be phosphorylated by cellular deoxycytidine kinase to gemcitabine diphosphate (dFdCTP) and gemcitabine triphosphate (dFdCTP), both of which can inhibit processes required for DNA synthesis. Other commonly used chemotherapies for pancreatic cancer include 5-fluorouracil (5-FU), oxaliplatin, albumin-bound paclitaxel, capecitabine, cisplatin, irinotecan, and docetaxel [15,16]. Although several gemcitabine-based combination treatments exist, most have not considerably improved survival. While some combinatorial chemotherapy treatments, such as gemcitabine with erlotinib, have demonstrated potential for longer patient survival, the majority of patients eventually experience tumor progression due to the development of resistance, and therefore novel therapies are required, especially those that do not rely solely on chemotherapeutic drugs [17,18].

The mechanisms of de novo or inherent resistance of PDACs to chemo- or radiotherapeutics are not well understood. Several factors have been demonstrated to contribute to such resistance, including (i) multiple factors associated with the nature of the PDAC tumor microenvironment (TME) [19,20]; (ii) nucleoside transporters or/and nucleoside enzymes affecting drug uptake and metabolism [21]; (iii) hypoxia-inducible factor-1 alpha (HIF-1α) regulated glucose metabolism [22]; (iv) stromal-derived Insulin-like Growth Factors (IGFs) [23]; (v) abnormal expression of tumor-associated mucin proteins [24]; (vi) IFN-related DNA-damage resistance signature (IRDS) of some tumors [25]. The understanding of chemoresistance of PDACs to chemotherapy is very important, as at least some of these mechanisms could be also contributing to the resistance of PDACs to OV therapy.

The success of any treatment for PDAC is further complicated by the TME of PDAC, which is characterized by dense stroma comprised of abundant fibroblasts, hypoxia, and sparse vasculature. Moreover, the infiltration of tumor-promoting immune cells mediates immune evasion and promotes tumor progression. The stroma surrounding the tumor is primarily composed of pancreatic stellate cells (PSCs) which are activated by secreted factors such as TNFα, TGF-β, and interleukins 1, 2, 10, and themselves secrete mucins, collagen, fibronectin, and laminin in addition to some other factors, forming a thick extracellular matrix (ECM). This composition generates an incredibly dense physical barrier, to both host immune cells and potential therapeutics while also increasing interstitial pressure, which, when combined with sparse vasculature, forms a hypoxic environment, further inhibiting immune cells in terms of recruitment and effectiveness. PI3K/Akt, a key downstream mediator of many receptor tyrosine kinase signaling pathways involved in cell proliferation, migration, and inhibition of apoptosis, is phosphorylated under hypoxic conditions, along with MAPK (Erk), which regulates cell proliferation in response to various growth factors, which have been associated with resistance to gemcitabine [26,27]. The limits on antitumor immune cell recruitment also leads to T-cell exhaustion resulting in loss of cytotoxic effector function and further limits appropriate immune responses. SDF-1α/CXCR4 signaling-induced activation of the intracellular FAK-AKT and ERK1/2 signaling pathways and a subsequent IL-6 autocrine loop in cancer cells can further increase chemoresistance [28].

The low expression of nucleoside transporters (NT) and inactivity of nucleoside enzymes (NE) both affect the activity of gemcitabine. Low expression of a nucleoside transporter hENT1 restricts the uptake of gemcitabine, preventing its incorporation into the DNA of replicating cancer cells, and high expression of hENT1 is related to longer overall survival in pancreatic cancer patients [29,30]. The inactivation of deoxycytidine kinase (dCK), an enzyme responsible for the initial phosphorylation of gemcitabine, also mediates resistance. dCK is often inactivated in gemcitabine-resistant PDAC lines [31], and knockdown of dCK has been shown to lead to the development of resistance [32], while expression of a DCK transgene (along with uridine monophosphate kinase) sensitized pancreatic cancer cells to gemcitabine [33].

Pancreatic cancers metabolize glucose at higher rates and show higher expression of HIF-1α positively correlated with gemcitabine resistance [34,35]. HIF-1α increases glucose uptake and metabolism in the cell and is stabilized by MUC1, a common biomarker for cancers including PDAC [36]. Knockdown of HIF-1α in gemcitabine-resistant cells reduced tumor cell survival following gemcitabine treatment, and treatment with digoxin, and HIF-1α inhibitor, reduced glucose uptake and cell survival in cells treated with gemcitabine [37]. The increased glucose uptake under hypoxic conditions feeds into the glycolysis pathway and increases biomass; however, the exact mechanisms by which HIF-1α reduces sensitivity to chemotherapeutics have yet to be determined.

In addition, stromal-derived IGFs activate the insulin/IGF1R survival signaling pathway, reducing responsiveness to chemotherapeutics [38]. One proposed mechanism describes crosstalk between activated Insulin/IGF signaling pathways in PDAC. IGF-1 and IGF-1R, which are known to be abundantly expressed in the PDAC tissue, can stimulate β-cell proliferation and increase β-cell mass, increasing basal insulin production which may alter the trophic effects of the endocrine cells on the exocrine cells. Endocrine β-cells that express oncogenic K-ras can also be one potential progenitor for PDAC under chronic tissue inflammation [39]. This is further supported by evidence that demonstrates macrophages and myofibroblasts are the two major sources of IGFs within the pancreatic tumor microenvironment, and that chemoresistance is increased when cytotoxic agents increase M2-like macrophage infiltration [23]. For any novel therapies to be effective, they should be able to address most if not all of these challenges.

The structural composition of mucins produced by cells in certain cancers, such as breast and pancreatic cancers, has been suggested to limit immune cell recognition by blocking infiltration [40]. Similarly, the dense mucin mesh prevents cellular uptake of chemotherapeutics like gemcitabine and 5-FU within the tumor. MUC1 and MUV4 are overexpressed and aberrantly glycosylated in the majority of pancreatic tumors [41]. Kalra et al. demonstrated that the inhibition of mucin *O*-glycosylation enhanced the cytotoxic effects of 5-FU against human pancreatic cancer cell lines, but not against the mucin-deficient cell line [40]. They suggest that preventing the formation of the mucin facilitates the diffusion of drugs across the compromised mucus layer, improving intracellular drug uptake and enhancing cytotoxic drug action. Elevated MUC1 and MUC4 expression have also been correlated with greater degrees of resistance to gemcitabine [42]. It was also demonstrated that gemcitabine-resistant cells had accentuated the non-oxidative branch of the pentose phosphate pathway activity and increased pyrimidine biosynthesis, conferring resistance by increased dCTP production. MUC1 and MUC4 overexpression was also shown to upregulate *mdr* genes in pancreatic cancer cells, including *ABCC1, ABCC3, ABCC5,* and ABCB1 genes [41,43]. MUC4 expression was shown to be conversely correlated with the expression of hCNT1 and hCNT3 transporters, preventing uptake of chemotherapeutic drugs like gemcitabine, and hCNT1 is upregulated when MUC4 is inhibited, resulting in increased drug sensitivity [44]. Finally, MUC4-overexpressing CD18/HPAF-Src were not sensitive to gemcitabine, conferring resistance and survival advantages through erbB2-dependent and anti-apoptotic pathways [45]. Altogether, mucins including MUC1 and MUC4 have been demonstrated to be highly overexpressed and aberrantly glycosylated in pancreatic cancer cells, conferring resistance to various chemotherapies and the downregulation of these oncoproteins may represent a promising therapeutic strategy for reversing chemoresistance and reducing tumor progression and mass.

Type I IFN signaling is upregulated in some tumors responding to chemotherapy and can have antitumor as well as pro-tumor effects. The expression of a type I IFN-related DNA-damage resistance signature (IRDS) was reported to correlate with resistance to chemotherapy and radiotherapy in multiple cancer types. In breast cancer, the IRDS has been implicated in the development of chemoresistance, which may be another potential mechanism of resistance in PDACs as well [25]. The STAT1/IFN pathway transmits a cytotoxic signal either in response to DNA damage or to IFNs, such as in the case of viral infection. Cells with an IRDS (+) profile show constitutive activation of the STAT1/IFN pathway. Interestingly, this chronically activated state of the STAT1/IFN pathway may select against transmission of a cytotoxic signal, instead resulting in pro-survival signals mediated by STAT1 and other IRDS genes [25]. In agreement with this mechanism, STAT1 is highly upregulated in many cancers, including PDAC, and protects SCC-61 cells from ionizing radiation-mediated death [46]. STAT1 may also induce resistance with other DNA damage-based treatments, such as gemcitabine, and may transduce survival/growth signals that enhance tumor survival under some conditions [47]. Sensitivity to DNA damage is coupled with sensitivity to IFNs such that selection for resistance to one may lead to resistance to the other [48], which could prove to be a problem with not only chemo- and radiotherapies, but OV treatments as well.

## 3. Overview of Common Experimental Models to Study OV Therapy in PDAC

Oncolytic virus (OV) therapy is a relatively novel anticancer approach. Effective OV therapy is dependent on the oncoselectivity of OVs—their ability to preferentially infect, replicate in and kill infected cancer cells without damaging nonmalignant (“normal”) cells. The ideal OV therapy not only requires the direct lysis of cancer cells by the virus but also activates innate and adaptive anticancer immune responses [49] (Figure 1).

Preclinical PDAC models are critical for understanding the biology of PDAC, are platforms for developing novel strategies against PDAC, and are a necessary part of the drug development pipeline. There are several features of an ideal PDAC model system to develop clinically relevant OV therapy against PDAC: (1) the ability to test OV against different PDACs, characterized by various responsiveness to different therapies, including OV therapy; (2) the model should recapitulate a complex TME of PDACs; (3) tractability of the model, including the ability to trace both tumor cells and OV; (4) the ability to deliver OV systemically, as the PDAC are difficult to access; (5) the ability to detect and evaluate innate and adaptive immune responses against both tumor cells and OV. Unfortunately, there is no single PDAC model that successfully recapitulates all these critical features and challenges of the disease. However, there are numerous models for PDAC, each with unique advantages and disadvantages. Here, we will briefly review the advantages and disadvantages of various in vitro and in vivo models of PDAC and how they can contribute to the development of OV therapeutics.

### 3.1. In Vitro Systems

#### 3.1.1. PDAC Cell Lines

Numerous human PDAC cell lines have been established and can be characterized by their distinctive genotypic and phenotypic variations, including their relative permissiveness or resistance to OV infection [50,51,52]. Utilizing cell lines as a model system offers several advantages for studying PDAC, including easy propagation and indefinite growth. These features represent a cost-effective and consistent model that can easily be used to study molecular mechanisms and biomarkers of resistance or permissiveness of PDAC cells to OVs [50,52]. While cell line-based approaches represent quick, straightforward, and consistent models, several features reduce their clinical translatability. First, the homogeneous nature of cell line models fails to accurately represent the heterogeneous nature of typical in vivo tumors, including PDAC [53]. Indeed, cell lines are under selection for mutations and phenotypes allowing growth advantage in a monolayer, however, the selection mechanisms in vivo are different [54]. In fact, established PDAC cell lines not only lose the heterogeneity present in the primary tumor, but the evolution of these cell lines to grow in culture may obscure genetic aberrations present in the primary tumor [52]. Additionally, many PDAC cell lines are originated from metastasized disease, so the ability to study PDAC progression is severely limited. Secondly, cell lines cultured in a monolayer lack the important three-dimensional structure and function as seen in vivo [54]. Thirdly, the PDAC cell line model fails to represent the TME, which is understood to be a dynamic player in PDAC tumor progression [54]. Lastly, cultured cell lines lack selection pressure from the host adaptive immune system, thus leaving mutations necessary for evading host immunity underrepresented. The outcome of the OV therapy depends on the complex interaction between tumor cells, virus, and innate and adaptive immune systems of the host. One of the desirable outcomes of this interaction is OV-mediated stimulation of immune response against tumor cells. However, normal PDAC stromal cells can induce innate antiviral responses against OV replicating in tumor cells, and adaptive immune response can prematurely clear virus infection instead of targeting tumor cells. Unfortunately, cell culture-based models cannot address these important issues.

Even given the disadvantages of the cell line model, it is a good starting proof-of-principle platform that has allowed our group to investigate mechanisms regarding responsiveness or resistance to OV therapy [50,51,55,56,57,58,59,60,61,62]. For example, our group is interested in understanding why/how certain PDAC cell lines are more resistant to VSV infection than other PDAC cell lines [50]. The cell line model in this aspect allows for reliable comparative measurements of virus replication, spread, and cell lysis. Additionally, the cell line model allows for relatively straightforward screening of both cellular and viral genes and proteins of interest. Cell line models allow for efficient virus tractability through reporter genes such as GFP [63]. Additionally, cell culture-based systems allow innovative imaging approaches for single-cell real-time analysis of OV replication and efficacy in pancreatic cancer cells [64].

Depending on the nature of the investigation, either human or murine PDAC cell lines can be used. Human PDAC cells, derived from primary pancreatic tumors or “cell line-derived xenograft (CDX)” models, have been used since as early as 1963 to characterize and test anti-cancer drugs [65]. The use of human PDAC cells provides the obvious benefit of having the same genetic makeup of the human disease, including key PDAC mutations in KRAS, CDKN2A, p53, and SMAD4 [3]. Although using human PDAC cell lines as a model has numerous informative applications, this model has a limited ability for consequent in vivo studies. If using human PDAC cell lines, researchers are limited to T cell-deficient nude athymic (nude), or B and T cell-deficient severe combined immunodeficient (SCID) mice [66,67]. As will be later discussed in this review, while such in vivo models have many applications, they lack the ability to assess the role of the adaptive immune system against PDAC as well as OV, both important when determining the efficacy of potential OV therapeutics.

To circumvent this caveat, murine PDAC cell lines can be used. Using murine PDAC cells derived from murine PDAC tumors allows researchers to establish PDAC in immunocompetent mice, allowing for the study of OV therapy in the presence of the functional adaptive immune system. One notable drawback to this model is the potential genetic dissimilarity (and thus clinical translatability) between mouse and human PDAC cells.

Generally, murine PDAC cell lines are originated from mice that have PDAC due to either chemical induction or genetic modifications in genetically engineered mouse model (GEMM). Once commonly used PDAC cell line that was cultured from a chemically induced PDAC tumor is Panc02, which has been extensively used for PDAC research [68]. The PDAC tumor from which it was derived was established by implanting 3-methyl-cholanthrene (3-MCA)-saturated threads of cotton in the pancreas of C57BL/6 mice. Despite its long-term use in evaluating various therapeutic strategies, Panc02 cells lack clinical significance for PDAC due to the absence of some common mutations found in human PDAC. More relevant murine PDAC cell lines are originated from the KPC mouse model of PDAC (LSL- Kras^G12D^; LSL-Trp53^R173H^; *Pdx*1-*Cre*) [68]. KPC mice develop spontaneous PDAC which closely resemble the genetics, physiology, tumor progression, and metastatic hallmarks of human PDAC [69], and will be described in more detail later in this review.

#### 3.1.2. PDAC Organoid Cultures

To better address the lack of 3D structure and function of 2D models, 3D organoid (organ-like) structures may be used that self-organize into structures that more closely resemble the in vivo tissue structure, composition, and function [70]. To model PDAC, normal or cancerous pancreatic ductal cells are typically embedded in Matrigel™, which contains important components of basement membrane and growth factors. Pancreatic ductal cells form polarized structures due to the cell–cell contacts and cell-matrix interactions, which can greatly influence gene expression when compared to 2D cultures [71]. The ability to better mimic the complex 3D architecture of PDAC in vitro is a valuable platform for testing drug delivery, pharmacokinetics, efficacy, and drug resistance. One of the key advantages of the organoid model is the ability to study PDAC progression by comparing normal pancreatic, preneoplastic, and PDAC cell-based organoids. Better understanding at what phases PDAC is more susceptible or resistant to OV therapy is of particular interest to our group, and such a model represents an unparalleled system for controlled disease progression and visibility. Although organoids have many promising potential applications, there are still limitations. First, this organoid model system is synthetic and the mutational selection is not well understood [72]. Second, many of these models are solely epithelial tissue layers, lacking important elements of the tumor microenvironment such as immune cells, nervous cells, mesenchyme, muscular layers, etc. [73]. This limitation is addressed in more realistic organoid models of PDAC, or, “PDACoids” are additionally co-cultured with stromal components like cancer-associated fibroblasts (CAFs), PSCs, endothelial cells, and immune cells to better mimic the dense stroma which typically represents up to 90% of the tumor volume and is a major player in PDAC tumor progression and therapeutic resistance [74]. Thirdly, the organoid model fails to address the complex immune system dynamics as in the human disease [75]. Unfortunately, even as organoid models continue to progress towards the true complexity of the in vivo tumor, the immune microenvironment around a tumor is exceedingly difficult to truly recapitulate in vitro. Despite the limitations of organoids, this technology has great potential and use to more closely model human tumors. We would like to refer to a more exhaustive review of 3D cell culture approaches, including the spheroid model systems here [76].

PDAC organoid cultures offer a more realistic model to study OV delivery, replication, cell lysis, and oncoselectivity, as they better mimic the 3D organization and complexity of the human disease. Recent studies showed that an adenovirus-based OV exhibited good oncoselectivity, with replication only occurring in organoids from PDAC tumors. The group also concluded that the cytotoxicity observed in PDAC organoids was predictive of antitumor efficacy in both subcutaneous (SC) and orthotopic xenograft models [77]. Although VSV has not yet been tested in a PDAC organoid setting, other OVs used so far [64,78,79] have been promising and provide a more predictive model for in vivo disease.

### 3.2. In Vivo Murine Model Systems

Murine models for human PDAC research are useful tools as mice and humans have comparable anatomic, cellular, and genomic features, including tumor biology [80]. For the scope of this review, we will focus on murine-based in vivo model systems, however in vivo PDAC models from alternative species are also used, and we refer to these excellent articles [81,82].

Murine models of PDAC can help both researchers and clinicians to better understand the onset, development, and metastatic processes of this disease, as well as to explore new therapeutic modalities such as OV therapy. Ideally, a murine model of PDAC should have the following features: (1) consistent PDAC disease progression similar to that of human disease from precursor lesions to PanIN and then PDAC [83]; (2) Cancerous phenotype similar to that in human disease demonstrating the common hallmarks such as anti-apoptosis, immune evasion and suppression, dense fibrosis/desmoplasia, and metastasis; (3) Should address the phenotypic and genetic heterogeneity as seen in human disease; (4) A reliable, consistent, and relatively quick time to tumor establishment; (5) Ability to study innate immune responses to PDAC cells as wells as OV; (6) Ability to track in vivo both tumor cells and OV. Here, we will briefly review the advantages and disadvantages of common in vivo PDAC models, and how they pertain to the development of OV therapeutics. We will break down these models by the genetic background of the mouse, and how PDAC is established in the mouse. For more comprehensive reviews of in vivo PDAC model systems, we refer to these papers [84,85].

#### 3.2.1. Human Cell Line Derived Xenograft (CDX) and Patient-Derived Xenograft (PDX) Models

Human CDX and PDX xenograft PDAC models are used by introducing PDAC cell lines (CDX) or primary tumor tissues (PDX) into immunocompromised mice (nude or SCID), commonly via SC injection [86]. These are useful models for studies not focused on antitumor immune responses, such as drug screenings as it is procedurally relatively simple and economical [87]. The SC CDX and PDX models have additional advantages: (1) the tumor has good tractability and is relatively easy to measure, even in the absence of reporter genes (e.g., luciferase); (2) depending on the growth rate of the cell line, tumors can be palpable within 2–6 weeks [85], and (3) this model allows for direct intratumoral injection of chemotherapeutics or OVs, and subsequent evaluation. However, these models have serious limitations for studying PDACs, which most often develop metastatic tumors, and SC tumors typically fail to metastasize. Furthermore, the CDX model is characterized by the loss of genetic heterogeneity in culture, whereas the PDX model at least retains some of the patients’ original genetic heterogeneity [88].

The orthotopic CDX and PDX xenograft PDAC models are more clinically relevant. In those models, PDAC cells or primary tumor tissues are injected/implanted into the pancreas of nude or SCID mice, which better recapitulates primary human tumors and are more likely to provide metastases and show more relevant tumor microenvironment compared to the SC model [89,90]. However, this approach is more procedurally challenging and requires special imaging techniques such as ultrasonography or an in vivo imaging system (IVIS) in concert with PDAC cells that express a reporter gene such as luciferase [67,89]. Moreover, ideally, the second reporter gene (e.g., for red fluorescent protein) should be encoded by OV to track virus spread in the tumors (primary and metastatic) and potential spread to normal tissues. The major limitation of all human CDX and PDX xenograft PDAC models (SC and orthotopic) is the lack of host immunity that both limits the study of OV-mediated adaptive antitumor and antiviral immunity and the robustness of the host, as immunocompromised mice typically exhibit susceptibility to infections and other health problems [91].

#### 3.2.2. Humanized Murine Model

The use of human CDX and PDX models are severely limited when investigating the dynamic interplay between the tumor, tumor microenvironment, and immune system while studying human PDACs. To compensate for this, researchers have developed humanized murine models, where mice are engineered to express components of the human immune system [92,93]. Humanized mice were created by establishing mutations in the IL2 receptor common chain (IL2rg^null^) in the non-obese diabetic (NOD)/SCID background [94,95]. With the combined lack of NK cell activity from the NOD background and the impaired B and T cell response from the SCID background, this model can support the implantation of human tissue, peripheral blood mononuclear cells (PBMCs) and hematopoietic stem cells (HSCs), allowing for the modeling of human adaptive immunity in immunocompetent mice [93,96]. It is important to note that, although these models allow for studies involving the adaptive immune system, these mice do not have the complete human immune system. This model has shortcomings such as limited lymph node development, HLA incompatibility between grafted human immune components and PDAC cells/tissue, and limited ability to mimic human immune cell trafficking [97]. Traditionally, human PDAC is characterized as non-immunogenic or “cold” tumors due to its lack of T-cell infiltration and immunosuppressive microenvironment [98]. However, tumor implantations in humanized murine models can cause T-cell infiltrations due to lack of histocompatibility, therefore changing classical cold tumors into artificially hot tumors, which can subsequently lead to false-positive results in immunotherapeutic investigations. Some studies have utilized the humanized murine model to try and better understand the role of the adaptive immune system and its role in anti-tumor immunity in the context of OV therapy [99]. While these studies may offer some insights into the role of anti-tumor immunity during OV therapy, there are many caveats to this model that still need to be addressed.

#### 3.2.3. Genetically Engineered Mouse Models (GEMMs)

The KPC Murine KPC cell lines provide a bridge between the need to recapitulate human PDAC disease phenotype and the use of immunocompetent in vivo murine models. This model was developed when they used a Cre/LoxP approach to express a mutant KRAS allele exclusively in pancreatic progenitor cells, causing the development of PanIN lesions which subsequently progressed to PDAC, but only after a prolonged latency period, as KRAS mutations alone are not sufficient for PDAC development [100]. This GEMM model mimics the classical characteristic of human PDAC, including the pronounced desmoplastic stromal reaction [100,101].

Since, numerous GEMM models have been developed to also include classical human PDAC mutations such as in TP53, SMAD4, CDKN2A, TGF-β, and INK4A. The details of these models are beyond the scope of this review, are more comprehensively described in other reviews [102,103,104]. As mentioned above, currently, the most commonly used model is LSL-KRAS^G12D^; LSL-Trp53^R172H^; PDX-1-Cre (KPC, stands for: Kras, p53, and Cre) mouse (C57BL6 genetic background). Unlike predecessor GEMM models, such as the LSL-KRAS^G12D^; PDX-1-Cre (KC) mouse, the KPC model creates advanced PDAC with classical human PDAC effects such as cachexia, abdominal distension, bowel and biliary obstruction. PDAC progression in KPC models closely resembles that of human disease as they develop PanIN within 8–10 weeks, followed by invasive and metastatic tumors by 16 weeks, along with the characteristic PDAC desmoplastic stromal reaction [69]. As the disease progresses, the tumor will predictably metastasize to the liver, lung, diaphragm and adrenals, as seen in the human disease. The KPC model, and GEMMs generally, provide the advantage of being able to investigate potential biomarkers, novel diagnostics, and/or therapeutics in the early stages of PDAC, and in the presence of the functional adaptive immune system [105,106]. The KPC model is an attractive platform for investigating the efficacy of OV therapies, due to the high semblance of PDAC disease and immune status of this model compared to that of the human disease.

This model does however have disadvantages. First, although resulting PDAC in the KPC model is very similar to that of the human disease, tumors are of murine origin and will therefore have inherent differences compared to human disease. Additionally, the development of KPC mice is labor-intensive, costly to upkeep, and tumor initiation and formation take up to or greater than one year [107]. As well, tractability is limited as monitoring tumor progression requires specialized equipment that might not be available to all labs [108]. These technical drawbacks of this model make it less than ideal for OV therapy testing.

As the vital and therapeutic role of the immune system continues to be acknowledged during OV therapy, it should be standard to use a murine model with a competent immune system. GEMMs, such as KPC mice, represent such a model but can be limited due to high costs, labor intensity, and long tumor formation periods. Syngeneic murine models are developed by introducing murine tumor cells or tissues into immunocompetent mice of the same or similar genetic background either SC or orthotopically, i.e., implanting PDAC cells or tissue from a C57BL6 background mouse into a “wild-type” (WT) C57BL6 mouse. One of the earliest murine PDAC cell lines cultured, Panc02, was established from chemically induced PDAC in 1984 [68]. However, due to the artificial induction from which these cells arose, they do not harbor classical mutations as in human disease, such as KRAS and p53 [109]. A promising alternative is the KPC cell lines, which originated from the KPC mice and containing clinically relevant genetic mutations [63]. Syngeneic murine models can be established in immunocompetent mice either orthotopically or SC, both having the unique advantages and disadvantages. Both approaches provide the important feature of exhibiting a full immune system. The process of injecting syngeneic PDAC cells SC is procedurally less laborious, and tumor tractability is good, but the SC approach lacks the overall clinical relevance compared to orthotopic due to the tumor not being in the pancreas, and its lack of reliable metastasis. The main limitation in the orthotopic approach is the lack of PDAC cell tractability. However, luciferase can be genetically engineered into the PDAC cell lines to be implanted, allowing for much easier tumor imaging by measuring intensity of bioluminescence [102]. Other methods of syngeneic cell line delivery include intravenous, intraperitoneal, and intrasplenic, which have been used to provide models for lung, peritoneal, lymph node, and liver metastasis, respectively [110,111,112]. Our group is currently developing the syngeneic KPC cell line model system for studying VSV-based OV therapy against PDAC (i.e., location, tumor microenvironment, immune system), without the high costs and long development times of GEMM models. A key highlight of this system is tractability of both tumor cells as well as VSV via encoded far-red fluorescent protein, and this system utilizes several alternative KPC cell lines that have been engineered to express luciferase, so tumor growth and subsequent metastasis can be tracked and measured easily [113,114].

In conclusion, different preclinical PDAC models provide platforms to study important aspects of PDAC tumor biology, and potential treatments. The in vitro PDAC cell line model allows for large-scale and/or high throughput screenings, as well as determining basic infectivity to OVs and innate immune status but lacks obvious physiological components such as a full immune system, tumor microenvironment, metastasis, and early progression. The in vitro organoid model shares many features of the PDAC cell line model, but better addresses tumor heterogeneity, tumor microenvironment, and disease progression. In vivo PDX models allow similar genetic representation of the human disease by using human-derived tissue, but lack major clinical features of PDAC such as early disease progression, complete immune system, and tumor microenvironment (if implanted SC), as PDXs must typically be implanted into an immunocompromised mouse. The humanized mouse model is a PDX alternative, better allows for studies of immune system interactions with both the tumor and tumor microenvironment, as well as potential immune-modulating treatments such as OV therapy. GEMMs, and in particular the KPC model, best recapitulate human PDAC in full, and are the most ideal systems for research purposes. Unfortunately, GEMM models are time-consuming and costly. The syngeneic model is far less time-consuming than the GEMM KPC model, and shares most of its beneficial features, with the exception of early disease progression studies. Each of these model systems have strengths and weaknesses, and the most suitable model depends on what questions are being asked. It is therefore important to understand the unique advantages and disadvantages of each PDAC model system. 

In this review, we will discuss current advances with OV therapy for PDAC, with a special focus on VSV, the major interest of our laboratory. While other OV will be discussed in the current review, we would like to refer to other reviews which give a more general overview of OV therapy for pancreatic cancer [115,116,117,118].

## 4. Overview of the Current Progress in OV Therapy for PDAC

In 2015, the FDA approved Talimogene laherparepvec (T-VEC; Imlygic™), a genetically modified herpes simplex virus, to treat melanoma [119]. T-VEC is the first and still the only FDA-approved OV. However, numerous OVs are currently in preclinical studies and clinical trials for various malignancies, including PDAC. Table 1 and Table 2 summarize the results of preclinical studies (Table 1) and clinical trials (Table 2) of various oncolytic viruses against pancreatic cancer.

Preclinical studies demonstrated that a wide range of different viruses could be efficient OVs against PDAC (Table 1).

Additionally, gene therapy targets, such as oncogene knockdown, insertion of functional tumor-suppressor genes, and expression of functional RNAs also demonstrate improved cancer-killing efficacy when combined with OV. One method uses adenoviruses and adeno-associated viruses to deliver apoptotic genes to tumor cells. Such gene therapy using Adenovirus subtype 5 mediates rat insulin promoter directed thymidine kinase (A-5-RIP-TK)/ganciclovir (GCV) gene therapy resulting in significantly enhanced cytotoxicity to both Panc1 and MiaPaCa2 pancreatic cancer cells in vitro [176]. Another review explored the potential use of OV expressing functional p53 [117]. Another method would use OV to deliver siRNA transgenes for oncogenetic knockdown, such as ONYX-411-siRNA^ras^ expressing a mutant K-ras siRNA which significantly reduced K-ras mRNA expression at 48 h posttreatment and improved oncolytic activity [134]. The inclusion of an endostatin-angiostatin fusion gene in VVhEA also showed significant antitumor potency in vivo [152].

There have been many experiments screening for more effective virotherapies within available libraries, and modulated viruses such as the adenovirus AdΔCAR-SYE has been shown to significantly suppress tumor growth, and complete regression of tumors was observed in vivo [129]. In addition, more efficient and tumor-specific targeting peptides and OV could be identified by using additional libraries, and modifications to existing OV based on these findings are also promising. Such modifications have been shown to be effective, with the adenoviruses VCN-01 variants ICOVIR-15K and ICOVIR-17 [130]. As discussed previously, the ability of the OV to modulate the ECM was observed, as tumors treated with VCN-01 showed a dramatic decrease in the intratumoral HA content [130]. Other adenoviral variants such as Ad5PTDf35(pp65) have also demonstrated T-cell stimulation and dendritic cell (DC) modulation to increase efficient transduction within a human context [133].

Combinatorial treatments of chemotherapies, or chemovirotherapy, such as OV paired gemcitabine, have demonstrated improved oncolytic capabilities in vitro and in vivo than either treatment on their own. In vitro and in vivo studies showed that myxoma virus (MYXV) and gemcitabine therapies can be combined sequentially to improve the overall survival in intraperitoneal dissemination (IPD) models of pancreatic cancer [177]. The addition of chemotherapies to OV therapy using a combination of an oncolytic herpes simplex virus-1 mutant NV1066 with 5-FU increased viral replication up to 19-fold compared with cells treated with virus alone, and similar results were achieved by the addition of gemcitabine [125]. Similarly, oVV-Smac combined with gemcitabine greater cytotoxicity and potentiated apoptosis [148]. H-1PV combined with cisplatin, vincristine or sunitinib induced effective immunostimulation via a pronounced DC maturation, better cytokine release and cytotoxic T-cell activation [154]. The addition of gene targets alongside chemovirotherapy has also shown greater cytotoxic efficiency, as with VV-ING4 in combination with gemcitabine [150]. Even in cell lines that demonstrate resistance to viral infection, resistance can be broken with simultaneous treatments. Viruses like VSV rely on nonspecific interactions with the cell surface during the earliest stages of infections, and polycations have been shown to improve viral production and increase oncolysis by increasing the amount of virus interacting with cells during attachment [60]. Additionally, the use of JAK inhibitors like ruxotinilib and IKK inhibitors like TPCA-1 have also been shown to increase viral reproduction and oncolysis [56,60]. Other potential combinatorial treatment regimens could include radiovirotherapy or chemoradiotherapy.

Some of these treatment methodologies are already being tested in clinical trials. Table 2 describes the OV currently being tested in clinical trials, and while some are still underway, OV including ONYX-15, AD5-yCD, and T-VEC are well-tolerated, and in some cases, biologically active, either alone or in combination chemovirotherapies [167,168,169,170,171,174].

## 5. Understanding Molecular Mechanisms of Responsiveness and Resistance of PDACs to VSV-Based OV Therapy

VSV is a prototypic nonsegmented negative-strand (NNS) RNA virus (order *Mononegavirales*, family *Rhabdoviridae*). VSV is a promising oncolytic virus against various malignancies, and it has several advantages as an OV [178,179,180]: (i) its basic biology and interaction with the host have been extensively studied. The oncoselectivity of VSV is mainly based on VSV’s high sensitivity to Type I interferon (IFN) mediated antiviral responses (and therefore inability to replicate in healthy cells), while it can specifically infect and kill tumor response cells, most of which lack effective Type I IFN responses; (ii) although WT VSV can cause neurotoxicity in mice, nonhuman primates, several VSV recombinants, including VSV-∆M51, have been generated which are not neurotropic but retain their OV activity; (iii) VSV has a broad tropism for different types of cancer cells (including PDACs), as its primary mode of entry into a host cell utilizes binding of the VSV-G protein to LDLR, which is ubiquitous, and VSV-G is also capable of using other common surface molecules for cell entry [179]; (iv) there is no preexisting immunity against VSV in most humans; (v) replication occurs in the cytoplasm without risk of host cell transformation; (vi) cellular uptake occurs rapidly; (vii) VSV has a small, easily manipulated genome, and novel VSV-based recombinant viruses can be easily engineered via reverse genetics to improve oncoselectivity, safety, oncotoxicity, and to work synergistically with host immunity and/or other therapies in a specific tumor environment (e.g., PDAC); (viii) as other members of the order *Mononegavirales*, and compared to positive-strand RNA viruses, VSV is less likely to mutate, and our recent study demonstrated long-term genetic stability of VSV recombinants carrying large transgenes [62]. All these and other advantages make VSV a promising candidate OV for PDAC treatment, and we have shown that VSV is effective against the majority of PDAC cell lines in vitro and in vivo [50,51]. Importantly, several phase I clinical trials using VSV against different malignancies are in progress (ClinicalTrials.gov for trials NCT03647163, NCT02923466, NCT03120624, NCT03865212, and NCT03017820).

VSV exhibits inherent oncotropism based largely on defective or reduced type I IFN responses, as specific genes associated with type I IFN responses are downregulated or functionally inactive [165,181]. In addition, IFN signaling can be inhibited by MEK/ERK signaling or by epigenetic silencing of IFN-responsive transcription factors IRF7 or IRF5 [182,183]. However, some PDACs do not have these defects and resist VSV infection like normal cells, which are sensitive to IFN-α treatment and capable of secreting type I IFNs following VSV infection [184].

There has been a demonstration of neurotoxicity in mice infected intranasally or intracranially, demonstrating a need for methods of improvement of VSV oncoselectivity and neurotropic safety without compromising oncolytic ability. There are at least eight approaches demonstrated to address these needs [178,179,180]: (i) mutating the VSV M protein; (ii) VSV-directed IFN-β expression; (iii) attenuation of VSV through disruption of normal gene order; (iv) mutating the VSV G protein; (v) introducing targets for microRNA from normal cells into the VSV genome; (vi) pseudotyping VSV; (vii) experimental adaptation of VSV to cancer cells; and (viii) using semi-replicative VSV. Most of the studies in our laboratory focus on VSV-∆M51 recombinants containing a deletion of the methionine residue at position 51 of the M protein, VSV-∆M51. This mutation results in an inability of VSV-M to inhibit nucleus-to-cytoplasm transport of cellular mRNA, including antiviral transcripts, in normal cells with functional antiviral signaling [185,186].

Our laboratory has characterized numerous human PDAC cells lines and discovered a wide range of susceptibility and permissiveness of different PDAC cell lines to VSV and other tested OVs [50,51,56,58,59,60,61]. The range includes “super-permissive” cell lines (such as MIA PaCa-2 and Capan1), “super-resistant” cell lines (such as HPAF-II, Hs766T), and well as many cell lines in between (such as SUIT2 and AsPC-1). Below we describe different mechanisms associated with the resistance of some PDACs to VSV.

### 5.1. Upregulated Type I IFN Signaling and Constitutive Expression of a Subset of IFN-Stimulated Genes (ISGs)

Our extensive analysis of a large number of human PDAC cell lines demonstrates that PDAC cell lines show surprising diversity with regard to their ability to produce and respond to type I IFNs, and the evaluation of IFN sensitivity and IFN-α and IFN-β production within a cell line may be used to predict its responsiveness to oncolytic treatment [50,51] (Figure 2).

Upregulated or residual expression of antiviral genes display four unique phenotypes (Figure 2): (i) no type I IFN production and not responsive to type I IFN, (ii) no type I IFN production but responsive to type I IFN, (iii) type I IFN production and responsive to type I IFN, (iv) super resistant PDACs: type I IFN production, responsive to type I IFN and constitutive expression of many antiviral IFN-stimulated genes (ISGs) [51,56,58].

We also conducted a transcriptome analysis to identify biomarkers for resistance of PDAC cell lines to VSV-ΔM51. Of the genes identified, six demonstrate constitutive co-expression in the VSV-resistant cell lines: MX1, EPSTI1, XAF1, GBP1, SAMD9, and SAMD9L [58]. Most of these genes are known to have an antiviral effect. Moreover, shRNA-mediated knockdown of MX1 showed a positive effect on VSV-ΔM51 replication in resistant PDAC cells, suggesting that at least some of the identified ISGs contribute to resistance of PDACs to VSV-ΔM51 [58]. Finally, we demonstrated that JAK inhibitors effectively break resistance to VSV-ΔM51 while affecting very few non-ISGs, suggesting that the constitutive expression of these genes is likely a causative factor for the phenotype of resistance [50,56]. Further evidence that host antiviral response to VSV-∆M51 infection is the source of resistance has been shown in infection with WT VSV, as even cell lines resistant to VSV-∆M51 are permissive to at least some degree to the WT-VSV, which is better able to evade antiviral responses in the host [50,51].

### 5.2. Role of Cell Cycle in Resistance of PDAC Cells to VSV

We have demonstrated that compounds inducing cell cycle arrest in G1/S-phase or S-phase strongly inhibited VSV-ΔM51 replication, while G_2_/M phase arrest dramatically enhanced the replication of VSV-ΔM51 in cells with functional antiviral signaling [61]. It was found that G2/M arrest strongly inhibited IFN production and expression of ISGs in response to exogenously added IFN. The replication of IFN-sensitive cytoplasmic viruses can be strongly stimulated during G2/M phase as a result of inhibition of antiviral gene expression, likely due to mitotic inhibition of transcription, a global repression of cellular transcription during G2/M phase. However, G_2_/M arrest did not stimulate the replication of VSV-ΔM51 in cells defective in IFN signaling, and it did not stimulate replication of WT VSV, which is more effective at evading antiviral responses [61]. Together, our study suggests that continuous cell cycle transition, a hallmark of cancer cells, could be another factor of oncoselectivity for many viruses, at it would facilitate viral replication via inhibition of antiviral responses in dividing cancer cells during G_2_/M phase. It also suggests that slowly dividing PDAC cells could be more resistant to some OVs than faster dividing PDACs.

### 5.3. Resistance to Virus-Mediated Apoptosis

Inhibition of apoptosis is a hallmark of many malignancies, including PDAC [187], and PDAC with decreased expression or activation of certain apoptotic proteins have the potential to limit/delay cell death following VSV infection [166,188,189,190]. Our study demonstrated that all three tested VSV recombinants (VSV-GFP (WT M gene), VSV-p1-GFP (WT M gene, GFP in the first position in the genome), and VSV-ΔM51-GFP) induced caspase 3 cleavage following infection, but VSV-ΔM51-GFP induced more caspase 3 cleavage in all cell lines with VSV-inducible Type I IFN responses, despite similar replication levels for the viruses [59]. This indicates a positive role for the ΔM51 mutation, and therefore host antiviral responses, in apoptosis induction, and is unlikely to be simply a result of virus attenuation as VSV-p1-GFP induced caspase cleavage similarly to VSV-GFP. Further, VSV-ΔM51-GFP induces both the extrinsic and intrinsic apoptosis pathways in most PDACs, however, we observed inhibition of VSV-induced and drug-induced apoptosis in some PDAC cells lines, and that was observed even when VSV replication was stimulated using JAK1 inhibitors [59]. In general, our study has demonstrated that resistance of some PDAC cell lines to VSV-mediated oncolysis could be not only due to type I IFN responses that limit virus replication, but also to cellular defects in apoptosis.

### 5.4. Inefficient Attachment of VSV to PDAC Cells

It is generally accepted that VSV tumor tropism is mainly dependent on the permissiveness of malignant cells to viral replication rather than on receptor specificity. However, when compared VSV attachment to different human PDAC cell lines, we observed a dramatically weaker attachment of VSV to HPAF-II cells, the most resistant human PDAC cell line [60]. Interestingly, although sequence analysis of low-density lipoprotein (LDL) receptor (LDLR) mRNA did not reveal any amino acid substitutions in this cell line, HPAF-II cells displayed the lowest level of LDLR expression and dramatically lower LDL uptake. We also showed that LDLR-independent attachment of VSV to HPAF-II cells and some other PDAC cell lines can be dramatically improved by treating cells with polybrene or DEAE-dextran [60].

## 6. Enhancing Responsiveness of PDAC Cells to Oncolytic Virotherapy with VSV

### 6.1. Combination of VSV with Small Molecule Inhibitors

As mentioned above, PDAC cell lines resistant to infection by VSV-ΔM51 demonstrate constitutive expression of numerous ISGs, most notably Mx1 [50,56]. Importantly, a similar expression profile (including upregulation of ISGs such as Mx1) and virus resistance phenotype was demonstrated for primary PDACs isolated from patients [176]. Treatment of resistant cell lines with JAK inhibitor I (a reversible inhibitor of JAK1, JAK2, JAK3 and TYK2) reduced ISG expression and partially overcame resistance to VSV suggesting potential for further improvement by utilizing other inhibitors and/or targeting additional pathways [50]. A similar enhancement of VSV replication was shown for ruxolitinib (JAK1/2 inhibitor), which was previously shown to break resistance of human head and neck cancer cells to VSV [56]. Interestingly, a similar strong inhibition of STAT1 and STAT2 phosphorylation, decreased expression of Mx1 and OAS, and stimulation of VSV-ΔM51 replication was also observed with TPCA-1, a known IKK-β inhibitor. Moreover, using an in situ kinase assay, we demonstrated that TPCA-1 can directly inhibit JAK1 kinase activity [56]. Thus, our study demonstrated that TPCA-1 is a unique dual inhibitor of IKK-β and JAK1 kinase.

### 6.2. Combination of VSV with Polycations

As mentioned above, some PDAC cell lines, including HPAF-II, are highly resistant to VSV due to a combination of a constitutive antiviral state and type I IFN-independent impaired VSV attachment. It was determined that the source of the impairment to attachment was not a result of mutations to the LDLR or its lowered expression levels, but rather by some other mechanism [60]. We have shown that treatment of cells with polycations improved LDLR-independent virus attachment, as both the cellular membrane and the viral envelope have negative net charges, and the polycations serve to counteract the repulsive electrostatic effects of the lipid barriers. Earlier studies showed a similar effect on cells treated with either DEAE-Dextran or polybrene before infection [191,192]. A novel combinatorial treatment with ruxolitinib and polycations demonstrated improved overall VSV replication and oncolysis and accelerated VSV replication kinetics compared to treatment with ruxolitinib only [60].

### 6.3. Combination of VSV with FDA-Approved Chemotherapeutic Drugs

VSV is inherently oncoselective due to its high sensitivity to type 1 IFN response, as research indicates that most cancer cells are defective in this type of antiviral signaling, and VSV-∆M51 is more sensitive than the WT virus, which is better able to inhibit these responses. G2/M arrest stimulates viral replication by inhibiting antiviral responses. Paclitaxel treatment stimulated the replication of VSV-ΔM51 and Sendai virus (another cytoplasmic NNS RNA virus that is also sensitive to type 1 IFN response) in multiple PDAC cell lines via inhibition of antiviral gene expression in treated cells [61]. In cells with functional type I IFN signaling, G2/M arrest inhibited the expression levels of type I and III IFNs, as well as inhibiting the upregulation of ISGs in response to the same amounts of exogenously added type I IFN [61]. A similar effect was also shown for colchicine [193].

### 6.4. VSV Encoding p53 Transgenes

Studies suggest that p53 enhances type 1 IFN signaling in normal cells and in some cancer cells types. However, our recent study demonstrated that VSV-encoded p53 trasngene actually inhibited antiviral signaling in PDAC cells, while stimulating VSV-ΔM51 replication in those cells [57]. Several potential reasons exist for the contrast to previous reports, including (1) differences between normal cells and different cancer cells in antiviral signaling, (2) constitutive activation of NF-κB pathway in a majority of PDACs, and (3) different timing and level of expression for VSV-encoded p53 in the cell lines used in this study. Future studies with VSVs expressing human p53 in animal models will determine if the oncoselective phenotype seen in PDAC cell lines is also observed in vivo, as it is necessary to see if the benefits are retained in an immunocompetent model without compromising safety.

### 6.5. Experimental Evolution of VSV

Replication-competent OVs can evolve under natural selection, and reversion to virulence or loss of oncolytic potential can threaten the safety and efficacy of virotherapeutic treatments. Evolutionary risk assessment studies are required to confirm safety, and can also serve the benefit of producing more potent OV, however, the most effective strategy is to combine rational design with evolution, allowing each engineered virus to mutate and fully adapt to its intended target cells. In our recent study, two VSV-ΔM51 variants containing human p53 were serially passaged on Suit-2 cells, resulting in mutations that adapted the viruses to better replicate in multiple PDAC cell lines without developing mutations in the p53 gene or losing oncoselectivity. The mutations of note that were acquired by the viruses include two separate mutations within the G protein sequence; both of these mutations achieved fixation within the span of the experiment in either virus and were identical. It was determined that the acquired G mutations stimulate VSV replication at least in part due to improved virus attachment to PDAC cells [62,194].

## 7. Future Directions and Conclusions

As discussed previously, several factors pose a series of challenges that will determine whether OV is suitable for cancer treatment, especially for PDAC, where any treatment is further complicated by the TME, which is characterized by dense stroma comprised of abundant fibroblasts, hypoxia, sparse vasculature, as well as infiltration of tumor-promoting immune cells mediating immune evasion and tumor progression. The ideal OV treatment should allow for sufficient delivery and penetration of PDACs with the virus, induction of adaptive antitumor responses, and prevention of premature OV clearance by host antiviral response. It is unlikely that any monotherapy could address all these challenges, and future effective OV-based treatments will likely be combinatorial (chemo-virotherapy, radio-virotherapy, chemo-radio-virotherapy, chemo-radio-immuno-virotherapy, etc.). Many of VSV-based combinatorial approaches have been described in our previously published reviews [178,180], and some additional approaches will be discussed below [195].

First of all, to study and address all these challenges, the ideal model systems should employ immunocompetent animals (to examine antiviral as well as antitumor immune responses) and be able to monitor not only tumor growth and spread, but also OV spread.

Figure 3 illustrates the system that we currently use in our laboratory to investigate various VSV-based OV treatments against PDAC.

PDAC is a highly heterogenic disease, and our studies have demonstrated dramatic differences between different human PDAC cell lines in their permissiveness to VSV and other OVs [50,51]. Future studies should define distinct subtypes of PDACs to develop personalized treatment strategies for different types of PDACs [3,196,197]. Although there is still no consensus classification for clinical application, some treatments work better against a particular PDAC subtype. For example, patients who had a germline BRCA mutation had significantly longer progression-free survival with maintenance a PARP inhibitor olaparib than with placebo [198]. It could be interesting to test combinations of OVs with olaparib against PDACs that have the BRCAness phenotype [10]. Interestingly, at least two studies showed the increased efficacy of OV therapy for thyroid carcinoma [199] and glioblastoma [200] when OV was combined with olaparib.

One of the major challenges for any PDAC treatment is insufficient drug delivery into the tumors because PDACs are hypovascular, densely packed with ECM components, have a high intratumoral tissue pressure, and very low tumor perfusion. Several previously developed approaches could be used to improve OV delivery into PDAC. For example, administration of a combination of cilengitide (angiogenesis inhibitor) and verapamil (Ca^2+^ channel blocker) promoted tumor angiogenesis, while improving gemcitabine delivery and therapeutic efficacy in mice [201]. Additionally, the angiotensin inhibitor losartan was shown to increase perfusion, drug and oxygen delivery [202]. A more recent study highlighted the potential importance of ROCK inhibition using the oral inhibitor fasudil for dual targeting of tumor tension and vasculature [203]. The administration of fasudil, a Rho-kinase inhibitor, and vasodilator, reduced intratumoral fibrillar collagen, improved sensitivity towards gemcitabine/nab-paclitaxel, and reduced metastasis formation on gemcitabine/abraxane treatment [203]. At least some of these drugs could potentially improve OV therapy when used in combination with VSV or other OVs.

The role of the stromal cells during OV therapy is still unclear, and it is likely dependent on the subtype of the particular PDAC. At least under certain conditions, the stromal cells could play a positive role during OV therapy by dampening antiviral responses within tumor and thus stimulating OV replication and OV-mediated oncolysis [204].

Other areas for development include approaches with a focus on antitumor immune stimulation. The TME of many cancers, including PDAC, is known to be immunosuppressive, due to various factors including a dense, fibrotic composition and a hypoxic environment that prevent access and activation of immune cells within the tumor [205,206]. Adoptive T-cell therapy augments the potency of T-cells by chaperoning virus into the tumor [120], overcoming the stromal barrier. Antigen-specific T-cells that were loaded with VSV-∆M51 can also be used to produce viral infection, replication, and subsequent oncolysis, as well as producing a proinflammatory environment that helped suppress the immunosuppressive nature of the TME. Immune tolerance mechanisms have been implicated as the main barrier to effective antitumor immunotherapy [207], and the natural flora of the gut has been indicated to possess the ability to exert influence over the immune response of the TME, resulting in immune tolerance that promotes tumor growth and development. Future experiments should examine the role of the natural flora in the efficacy of OV therapy for PDAC.

## Figures and Tables

**Figure 1 cancers-13-01171-f001:**
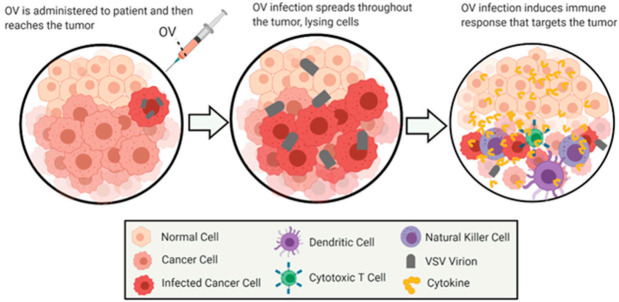
General Overview of Oncolytic Virotherapy. This figure demonstrates the general method of action for the treatment of cancer by oncolytic virotherapy using VSV as an oncolytic virus. The images depict the infection and oncolysis of malignant cells over time, followed by immunostimulation of cells invading the cleared area. The figure was created by authors with BioRender software (BioRender.com).

**Figure 2 cancers-13-01171-f002:**
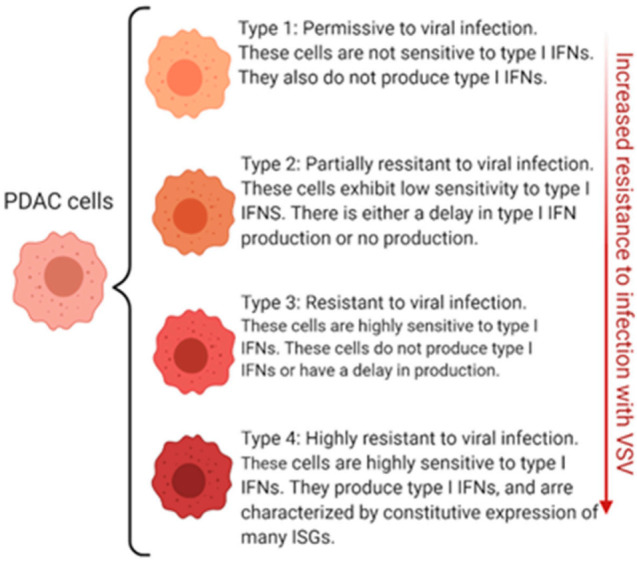
Permissiveness of PDAC to VSV: Four Different Phenotypes. This figure demonstrates the variability across PDAC in regard to permissiveness to infection by VSV. Permissiveness refers to the cells allowance for viral attachment, infection, and replication. The figure was created by authors with BioRender software (BioRender.com).

**Figure 3 cancers-13-01171-f003:**
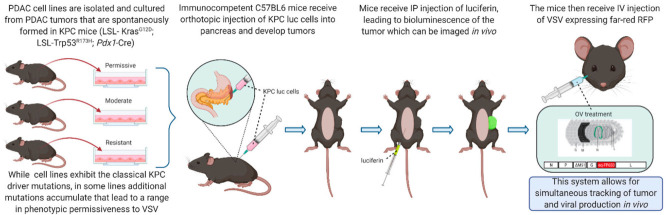
Syngeneic Model of PDAC in Mice. This figure shows the method of development for the murine model of PDAC. In this model, C57BL6 mice are used both for the development of the tumors and to evaluate the treatment in vivo. The figure was created by authors with BioRender software (BioRender.com).

**Table 1 cancers-13-01171-t001:** Preclinical studies of oncolytic viruses against pancreatic cancer.

Oncolytic Virus Backbone	Oncolytic Virus Name	Brief Description of the Results of the Study
Herpes virus	HSV-GFP (expressing NeoR and EGFP)	The pancreatic cancer lines MIAPACA and PANC-1 exhibited definite cytopathic effects upon infection in vitro (hPDAC cell lines) [120].
	HSV2 L1BR1 (US3-deficient)	US3-deficient HSV virus L1BR1 demonstrates a favorable characteristic regarding the induction of apoptosis in vitro (hPDAC cell lines) [121].
	HSV-1 (DF3gamma134.5)	The DF3/MUC1 promoter is shown to enhance the oncolytic activity of HSV-1 mutants in vitro (hPDAC cell lines) [122].
	R3616: γ134.5 hrR3: UL39	In vivo evaluation of two herpes virus mutants in combination with gemcitabine show complex interactions that can benefit or inhibit oncolytic activity (hPDAC IP into BALB/C nude, immunodeficient) [123].
	FusOn-H2	FusOn-H2 has potent activity against human pancreatic cancer xenografts in vivo (hPDAC SC or OT into Hsd nude, immunodeficient) [124].
	NV1066	5-FU and gemcitabine determined in vivo to potentiate oncolytic herpes viral replication and cytotoxicity across a range of clinically achievable doses in the treatment of human pancreatic cancer (hPDAC cell lines) [125].
	HSV(GM-CSF)	Injection of the recombinant mouse HSV encoding GM-CSF resulted in a significant reduction in tumor growth (mPDAC cells SC into C57BL6, immunocompetent) [126].
	Ad-DHscIL12	Expression of IL-12 in the context of a hypoxia-inducible oncolytic adenovirus is effective against pancreatic cancer in vivo (hamPDAC cells SC into nude, immunodeficient) [127].
	NV1020 G2O7	NV1020 and G2O7 effectively infect and kill human pancreatic cancer cells in vitro and in vitro (hPDAC cell lines) [128].
Adenovirus	AdΔCAR-WT, AdΔCAR-SYE, AdΔCAR-IVR	An in vitro tumor-targeting strategy using an adenovirus library for optimization of oncolytic adenovirus therapy (hPDAC cells SC into BALB/C nude, immunodeficient) [129].
	VCN-01	Oncolytic adenovirus VCN-01 shows an efficacy-toxicity profile in vivo (hPDAC cells SC into BALB/C nude, immunodeficient) [130].
	AdDeltaE1B19K dl337	Novel adenoviral mutants demonstrate the ability to improve efficacy of DNA-damaging drugs such as gemcitabine in vitro and in vivo (hPDAC cells SC into ICRF nude, immunodeficient) [131].
	WtΔE3ADP-Luc, WtΔE3ADP-IFN	Adenoviral death protein and fiber modifications significantly improved oncolysis in vitro and in vivo (hPDAC cells SC into NCr nude, immunodeficient) [132].
	Ad5PTDf35	This vector shows dramatically increased transduction capacity of primary human cell cultures including T cells, monocytes, macrophages, dendritic cells, pancreatic islets and exocrine cells, mesenchymal stem cells and tumor initiating cells (hPancreatic islet cells) [133].
	ONYX-411	These findings indicate that Internavec can generate a two-pronged attack on tumor cells through oncogene knockdown and viral oncolysis, resulting in a significantly enhanced antitumor outcome in vitro and in vivo (hPDAC cells SC into nude, immunodeficient) [134].
	ZD55-lipocalin-2	ZD55-lipocalin-2 may serve as a potent anticancer drug for pancreatic cancer therapy, especially for patients who have pancreatic adenocarcinoma with KRAS mutations as demonstrated in vitro (hPDAC cell lines) [135].
	LOAd703	LOAd703 is a potent immune activator that modulates the stroma to support antitumor responses in vitro and in vivo (hPDAC cells SC into C57BL/6 nude, immunodeficient) [136].
	OAd-hamIFN	Combination treatment of chemoradiation with IFN-expressing OAd demonstrates enhanced cancer-killing efficacy in vitro and in vivo (hamPDAC cells SC into Golden Syrian hamsters, immunocompetent) [137].
	OAd-TNFa-IL2	Ad-mTNFa-mIL2 increased immune cell infiltration to the tumor and altered host tumor immune status in vivo (hPDAC cells SC into NSG, immunodef and mPDAC cells SC into C57BL/6, immunocompetent) [138].
	YDC002	YDC002 combined with gemcitabine significantly attenuated the expression of major ECM components including collagens, fibronectin, and elastin in tumor spheroids and xenograft tumors compared with gemcitabine alone, resulting in potent induction of apoptosis, gemcitabine-mediated cytotoxicity, and an oncolytic effect through degradation of tumor ECM in vivo (hPDAC cells SC into BALB/C nude, immunodeficient) [139].
	AdΔΔ	AdΔΔ has low toxicity to normal cells while potently sensitizing pancreatic cancer cells to DNA-damaging drugs in vivo (hPDAC cells SC into C57BL/6 nude, immunodeficient) [140].
	dl922-947	dl922-947 is effectively able to elicit an anti-tumoral response in vivo when combined with 5-FU or gemcitabine (hPDAC cells SC into C57BL/6 nude, immunodeficient) [141].
	Delta-24-RGD	Delta-24-RGD significantly inhibited tumor growth in combination with phosphatidylserine targeting antibody in vivo (hPDAC cells SC into nude, immunodeficient) [142].
	Ad5-3Δ-A20T	Ad5-3Δ-A20T is highly selective for αvβ6 integrin-expressing pancreatic cancer cells for improved targeting of pancreatic cancer in vitro (hPDAC cell lines, 3D culture) [143].
	ICOVIR15	Arming the oncolytic adenovirus ICOVIR15 with miR-99b or miR-485 enhances its fitness and its antitumoral activity in vitro (human lung, breast, colorectal, prostate cancer cell lines) [144]
	Ad5-yCD/mutTK(SR39)rep-ADP	Ad5-yCD/mutTK(SR39)rep-ADP improves oncolysis in vitro and in vivo in combination with radiotherapy (hPDAC cells IM into CD-1 athymic, immunodeficient) [145].
Myxoma Virus	vMyxgfp	vMyxgfp had the ability to infect all pancreatic cancer cell lines tested in vitro (hPDAC cell lines) [146].
Reovirus	Reolysin	Reolysin treatment stimulated selective reovirus replication and decreased cell viability in KRas-transformed immortalized human pancreatic duct epithelial cells and pancreatic cancer cell lines in vitro *and* in vivo (hPDAC cells SC into nude, immunodeficient) [147].
Vaccinia Virus	oVV-Smac	oVV-Smac is indicated to have a synergistic effect in combinatorial treatment with gemcitabine in vitro and in vivo (hPDAC cells SC into BALB/C nude, immunodeficient) [148].
	VV-HBD2-lacZ	These results indicate that HBD2-expressing VV recruited plasmacytoid DCs (pDCs) to the tumor location, leading to cytotoxic T cell response against the tumor, and thus inhibited tumor growth in vitro and in vivo (murine melanoma cells SC into C57BL/6, immunocompetent) [149].
	GLV-1h68	GLV-1h68 was able to infect, replicate in, and lyse tumor cells in vitro and in vivo (hPDAC cells SC into BALB/C nude, immunodeficient) [150].
	VVLΔTK-IL-10	VV expressing IL-10 demonstrates enhanced anti-tumor efficacy in vivo (GEMM (KPC), immunocompetent) [151].
	VVhEA	The novel Lister strain of vaccinia virus armed with the endostatin-angiostatin fusion gene displayed inherently high selectivity for cancer cells, sparing normal cells both in vitro and in vivo, with effective infection of tumors (hPDAC cells SC into BALB/C nude, immunodeficient) [152].
Parvovirus	H-1PV	H-1PV in combination with gemcitabine enhanced anti-tumor activity of NK cells and effects included reduction in tumor growth, prolonged survival of the animals, and absence of metastases on CT-scans in vitro (hPDAC cell lines) [153].
	H-1PV	In ex vivo human models, H-1PV reinforced drug-induced tumor cell killing and effective immunostimulation (human melanoma cell lines) [154].
	H-1PV	The combination treatment of H-1PV and histone deacetylase inhibitors (HDACIs) such as valproic acid (VPA)acts synergistically to kill a range of human cervical carcinoma and pancreatic carcinoma cell lines by inducing oxidative stress, DNA damage and apoptosis in vitro and in vivo (hPDAC cells SC into NOD/SCID nude, immunodeficient) [155].
Measles Virus	MV-PNP-anti-PSCA	PNP, which activates the prodrug fludarabine effectively, enhanced the oncolytic efficacy of the virus on infected and bystander cells in vitro and in vivo (hPDAC cells SC into NOD/SCID nude, immunodeficient) [156].
	MeV	The chemovirotherapeutic combination of gemcitabine plus oncolytic MeV resulted in improved tumor reduction in vitro (hPDAC cell lines) [157].
Newcastle Disease Virus	NDV	NDV infection was successful in all evaluated PA cell lines in vitro, however the resultant replication kinetics and cytotoxic effects differed (hPDAC cell lines) [158].
	NDV	Infection with NDV activated immune cells which successfully elicited an anti-tumor response in vitro. However, activated NK cells that are abundant in Panc02 tumors lead to outgrowth of nonimmunogenic tumor cells with inhibitory properties (hPDAC cells OT into C57BL/6, immunocompetent) [159].
	MTH-68/H	MTH-68/H selectively kills tumor cell cultures in vitro by inducing endoplasmic reticulum stress leading to p53-independent apoptotic cell death (hPDAC cell lines) [160].
Poxvirus	CF33	CF33 caused rapid killing of six pancreatic cancer cells lines in vitro, releasing damage-associated molecular patterns, and regression of tumors in vivo (human colorectal cells SC into Hsd nude, immunodeficient) [161].
Influenza Virus	PR8, H5N1, H7N3, H4N8, H7N7, H5N1 HP, H7N1 HP	IAV significantly inhibited tumor growth following intratumoral injection without inducing apoptosis in nonmalignant cells in vivo (hPDAC cells SC into SCID, immunodeficient) [162].
Rhabdovirus	M51R-VSV	M51R-VSV treatment appears to induce antitumor cellular immunity in vivo (hPDAC cells SC into C57BL/6 nude, immunodeficient) [163].
	VSV-FH	VSV-FH can induce potent oncolysis in hepatocellular and pancreatic cancer cell lines in vivo (hPDAC cells SC into athymic nude, immunodeficient) [164].
	VSV-ΔM51	VSV showed oncolytic abilities superior to those of other viruses, and some cell lines that exhibited resistance to other viruses were successfully killed by VSV in vitro (hPDAC cell lines) [51].
	VSV-mp53, VSV-ΔM-mp53	VSV expressing p53 exhibited enhanced oncolytic action, while VSV-ΔM-mp53 was extremely attenuated in vivo due to p53 activating innate immune genes (hPDAC cells IV into BALB/C nude, immunodeficient) [165].
	VSV-ΔM51	VSV recombinants induced robust apoptosis in cells with defective IFN signaling, however cell lines constitutively expressing high levels of IFN-stimulated genes (ISGs) were resistant to apoptosis even when VSV replication levels were dramatically increased by Jak inhibitor I treatment in vitro (hPDAC cell lines) [61].
	VSV-WT, VSV-rM51R-M	Recombinant M51R-M (rM51R-M) virus induces apoptosis much more rapidly in L929 cells than viruses expressing WT M protein by a distinct method in vitro (murine fibroblast cell lines) [166].
	VSV-ΔM51	TPCA-1 (IKK-β inhibitor) and ruxolitinib (JAK1/2 inhibitor), as strong enhancers of VSV-ΔM51 replication and virus-mediated oncolysis in all VSV-resistant cell lines in vitro (hPDAC cell lines) [56].
	VSV	Combining VSV with ruxolitinib and Polybrene or DEAE-dextran successfully broke the resistance of HPAF-II cells to VSV by simultaneously improving VSV attachment and replication in vitro (hPDAC cell lines) [60].
	VSV, VSV-GFP, VSV-ΔM51-GFP	In vivo administration of VSV-ΔM51-GFP resulted in significant reduction in tumor growth for tested mouse PDA xenografts and antitumor efficacy was further improved when the virus was combined with gemcitabine (mPDAC cells SC into C57BL/6, immunocompetent) [55].
	VSV-p53wt, VSV-p53-CC	Two independently evolved VSVs obtained identical glycoprotein mutations, K174E and E238K; these acquired G mutations improved VSV replication, at least in part due to improved virus attachment to SUIT-2 cells, as determined in vitro (hPDAC cell lines) [62].

hPDAC = human PDAC, mPDAC = mouse, SC = subcutaneous injection; OT = orthotopic injection, IP = intraperitoneal injection; IV = intravenous injection.

**Table 2 cancers-13-01171-t002:** Clinical trials featuring oncolytic viruses against pancreatic cancer.

Oncolytic Virus Backbone	Oncolytic Virus Name	Brief Description of the Clinical Trial
Adenovirus	ONYX-015 (dl1520)	ONYX-015 injection via EUS into pancreatic carcinomas by the transgastric route with prophylactic antibiotics is feasible and generally well tolerated either alone or in combination with gemcitabine [167].
	ONYX-015 (dl1520)	Intratumoral injection of an E1B-55 kDa region-deleted adenovirus into primary pancreatic tumors was feasible and well-tolerated at doses up to 10(11) PFU (2 x 10(12) particles), but viral replication was not detectable [168].
	Ad5-yCD/mutTKSR39rep-ADP	A combination of intratumoral Ad5-DS and gemcitabine is safe and well tolerated in patients with LAPC [169].
	Ad5-yCD/mutTKSR39rep-hIL12	Ongoing clinical trial, no results posted to date. NCT03281382.
	LOAd703	Ongoing clinical trial, no results posted to date. NCT02705196.
	VCN-01	Ongoing clinical trial, no results posted to date. NCT02045589.
	VCN-01	Ongoing clinical trial, no results posted to date. NCT02045602.
Herpesvirus	T-VEC	EUS-guided FNI of T-VEC in advanced pancreatic ca, at initial doses of 104 to 106 PFU/mL followed by up to 107 PFU/mL, was feasible and tolerable. Evidence of biologic activity was observed [170].
	T-VEC	Ongoing clinical trial, no results posted to date. NCT03086642.
	HF10	HF10 direct injection under EUS-guidance in combination with erlotinib and gemcitabine was a safe treatment for locally advanced pancreatic cancer [171].
	HF10	Ongoing clinical trial, no results posted to date. NCT03252808.
	OrienX010	Ongoing clinical trial, no results posted to date. NCT01935453.
Reovirus	Reolysin	Pelareorep was safe but ineffective when administered with carboplatin/paclitaxel, regardless of KRAS mutational status. Immunologic studies suggest that chemotherapy backbone improves immune reconstitution and that targeting remaining immunosuppressive mediators may improve oncolytic virotherapy [172].
	Reolysin	PD analysis revealed reovirus replication within pancreatic tumor and associated apoptosis. Upregulation of immune checkpoint marker PD-L1 suggests future consideration of combining oncolytic virus therapy with anti-PD-L1 inhibitors [173].
	Reolysin	Pelareorep and pembrolizumab added to chemotherapy did not add significant toxicity and showed encouraging efficacy [174].
	Reolysin	Ongoing clinical trial, no results posted to date. NCT01280058.
Parvovirus	ParvOryx	The drug was safe and well-tolerated and showed a promising profile of anti-tumor effects and signs of clinical efficacy, i.e., prolonged survival. However, the optimum dose as well as the most appropriate route and schedule of administration have to be further investigated [175].

PFU = plaque-forming unit.

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
