# Peer review of "Expanding the Spectrum of Pancreatic Cancers Responsive to Vesicular Stomatitis Virus-Based Oncolytic Virotherapy: Challenges and Solutions"

_cancers, 2021, doi:10.3390/cancers13051171_

Round 1

Reviewer 1 Report

In this review, Hollbrook et al describes the use of oncolytic virus such as vesicular stomatitis virus (VSV) for the therapy of pancreatic cancer (PDC) a disease with no cure.

The review is well-written and the illustrations are adequate. This review should be definitely accepted for publication, as this is a great contribution to the field. However, one question remains… Why this virus and not another one? Here, I’m not speaking of intrinsic properties of VSV such as safety, delivery… but rather what makes pancreatic tumor a specific cellular/molecular playground for this viral strain? The authors should better elaborate on this.

Minor points and complements: there are several interesting missing references from the groups of Kleeff et al (PDAC biology), Buscail et al (Gene therapy, KRAS and PDAC biology) and Vonderheide (immunotherapy for PDAC). The work with Olaparib for patient with BRCAness profile should also be cited. There are interesting gene therapy trials aimed at restoring DCK expression in PDAC tumors (again by Buscail et al). This should be cited in the text just following reference #26. Regarding PDAC models, a recent publication by Quillien et al in HGT demonstrate the oncolytic effect of a novel myxoma strain in primary cells from patient, hence cultured as tumoroids. This should be added in the in vitro systems section. Last, the work of Carolina Ilkow (Nat Med 2015) on the impact of the tumor stroma on viral infection must be discussed, especially in the context of PDAC.

Author Response

REVIEWER 1:

GENERAL COMMENT: The review is well-written and the illustrations are adequate. This review should be definitely accepted for publication, as this is a great contribution to the field.

OUR RESPONSE: We thank Reviewer 1 for positive comments and excellent suggestions.

COMMENT 1: Why this virus and not another one? Here, I’m not speaking of intrinsic properties of VSV such as safety, delivery… but rather what makes pancreatic tumor a specific cellular/molecular playground for this viral strain? The authors should better elaborate on this.

OUR RESPONSE: We agree that needs to be explained, and it is now added to the revised manuscript.

COMMENT 1: there are several interesting missing references from the groups of Kleeff et al (PDAC biology), Buscail et al (Gene therapy, KRAS and PDAC biology) and Vonderheide (immunotherapy for PDAC). The work with Olaparib for patient with BRCAness profile should also be cited. There are interesting gene therapy trials aimed at restoring DCK expression in PDAC tumors (again by Buscail et al). This should be cited in the text just following reference #26. Regarding PDAC models, a recent publication by Quillien et al in HGT demonstrate the oncolytic effect of a novel myxoma strain in primary cells from patient, hence cultured as tumoroids. This should be added in the in vitro systems section. Last, the work of Carolina Ilkow (Nat Med 2015) on the impact of the tumor stroma on viral infection must be discussed, especially in the context of PDAC.

OUR RESPONSE: Thank you for pointing to these important papers. As suggested, all these papers are incorporated now into the revised manuscript.

Reviewer 2 Report

Th authors have done a very good to review of the field of studies of pancreatic tumor using oncolytic VSV.  They discussed the challenges and solutions. They have provided a long, long overview of common experimental models to study OV therapy in PDAC (in 8 pages!).  They have also discussed extensively molecular mechanisms of responsiveness and resistance of PDACs to oncolytic VSV therapy.

There is one major issue: The title of the review article is “Expanding the spectrum of pancreatic cancers responsive to oncolytic virotherapy: challenges and solutions”.  However, the manuscript is focused exclusively on Vesicular stomatitis virus (VSV)-based oncolytic virus. The only places touching other types of oncolytic viruses are Tables 1 and 2, where preclinical and clinical studies of various oncolytic viruses against pancreatic cancer are listed. In this case, other OVs serve little more than a laundry list, as they have not been explained in the main text at all.  Therefore, the title has to reflect the real content and needs to be modified.  It can be probably modified to, “Expanding the spectrum of pancreatic cancers responsive to VSV-mediated oncolytic virotherapy: challenges and solutions”. 

Minor points are,

  1. A few typos have been noticed. For example, Line 50: “KRAS2”?
  2. References: Some e-journals have so called “article number” to replace the traditional page numbers for citation purpose. These include most journals from the MDPI Publisher (Cancers, Biomedicines, Viruses, etc). Unfortunately, their article numbers cannot be downloaded into the library file when using reference management software (such as Endnote). You have to enter the article number into the file manually, in most cases. For this manuscript, there is a long list of references with these minor issues (mostly for article numbers, but some other items in other cases):

Ref #7, #11 (other error); 21 (volume and page numbers); 41; 54; 55; 56; 71; 78; 96; 110; 113; 127 (name of the journal?); 132; 139; 170; and 195.

Author Response

GENERAL COMMENT:  The authors have done a very good to review of the field of studies of pancreatic tumor using oncolytic VSV.  They discussed the challenges and solutions. They have provided a long, long overview of common experimental models to study OV therapy in PDAC (in 8 pages!).  They have also discussed extensively molecular mechanisms of responsiveness and resistance of PDACs to oncolytic VSV therapy.

OUR RESPONSE: We thank Reviewer 2 for positive comments and excellent suggestions.

COMMENT 1: There is one major issue: The title of the review article is “Expanding the spectrum of pancreatic cancers responsive to oncolytic virotherapy: challenges and solutions”.  However, the manuscript is focused exclusively on Vesicular stomatitis virus (VSV)-based oncolytic virus. The only places touching other types of oncolytic viruses are Tables 1 and 2, where preclinical and clinical studies of various oncolytic viruses against pancreatic cancer are listed. In this case, other OVs serve little more than a laundry list, as they have not been explained in the main text at all.  Therefore, the title has to reflect the real content and needs to be modified.  It can be probably modified to, “Expanding the spectrum of pancreatic cancers responsive to VSV-mediated oncolytic virotherapy: challenges and solutions”. 

OUR RESPONSE: Great suggestion! We have modified our title to: “Expanding the spectrum of pancreatic cancers responsive to vesicular stomatitis virus-based oncolytic virotherapy: challenges and solutions”

COMMENT 2: Minor points are: 1: A few typos have been noticed. For example, Line 50: “KRAS2”?

OUR RESPONSE: Corrected!

COMMENT 2: Minor points are: 2: References: Some e-journals have so called “article number” to replace the traditional page numbers for citation purpose. These include most journals from the MDPI Publisher (Cancers, Biomedicines, Viruses, etc). Unfortunately, their article numbers cannot be downloaded into the library file when using reference management software (such as Endnote). You have to enter the article number into the file manually, in most cases. For this manuscript, there is a long list of references with these minor issues (mostly for article numbers, but some other items in other cases): Ref #7, #11 (other error); 21 (volume and page numbers); 41; 54; 55; 56; 71; 78; 96; 110; 113; 127 (name of the journal?); 132; 139; 170; and 195.

OUR RESPONSE: All corrected!

Reviewer 3 Report

This is an extensive review of the advantages and disadvantages of experimental systems used to study pancreatic ductal carcinoma (PDAC) coupled with a review of studies of oncolytic viruses and PDAC, particularly vesicular stomatitis virus (VSV), the area of the authors' research. This will be a very useful contribution to those interested in this field, and is very thorough in these two main areas of emphasis.  Areas for suggested revisions are the following:

  1. Line 145: Instead of "overall survival" (usually applied to clinical studies), it should be "tumor cell survival following gemcitabine treatment".

  1. Lines 175-185: This discussion is very confusing, switching back and forth between MUC1 and MUC4.

  1. Several experimental approaches are described as "easy", when in fact they require some expertise in the area. Instead, describe them as "straightforward", etc.

  1. Lines 423-436: Needs editing.

  1. Lines 549-559: Many abbreviations need definition (MYXV, IPD, NV1066, etc.).

  1. Sections 5.4 and 6.2 say essentially the same thing.

  1. Similarly, the authors' results with JAK inhibitor are mentioned often in the manuscript.

  1. Line 779: Define SFN.

  1. Line 844: Define NRRP

Author Response

GENERAL COMMENT:  this is an extensive review of the advantages and disadvantages of experimental systems used to study pancreatic ductal carcinoma (PDAC) coupled with a review of studies of oncolytic viruses and PDAC, particularly vesicular stomatitis virus (VSV), the area of the authors' research. This will be a very useful contribution to those interested in this field, and is very thorough in these two main areas of emphasis. 

OUR RESPONSE: We thank Reviewer 3 for positive comments and excellent suggestions.

COMMENT 1: Line 145: Instead of "overall survival" (usually applied to clinical studies), it should be "tumor cell survival following gemcitabine treatment".

OUR RESPONSE: We agree with this comment – corrected!

COMMENT 2: Lines 175-185: This discussion is very confusing, switching back and forth between MUC1 and MUC4.

OUR RESPONSE: Corrected!

COMMENT 3: Several experimental approaches are described as "easy", when in fact they require some expertise in the area. Instead, describe them as "straightforward", etc.

OUR RESPONSE: Corrected!

COMMENT 4: Lines 423-436: Needs editing.

OUR RESPONSE: We

COMMENT 5: Lines 549-559: Many abbreviations need definition (MYXV, IPD, NV1066, etc.).

OUR RESPONSE: Defined!

 COMMENT 6: Sections 5.4 and 6.2 say essentially the same thing.

OUR RESPONSE: The section 5.4 is shorten to remove this redundancy

 COMMENT 7: Similarly, the authors' results with JAK inhibitor are mentioned often in the manuscript.

OUR RESPONSE: We removed most of these redundancies.

 COMMENT 8: Line 779: Define SFN.

OUR RESPONSE: Defined!

 COMMENT 9: Line 844: Define NRRP

OUR RESPONSE: Defined!